# Transcriptome Analysis of Improved Wool Production in Skin-Specific Transgenic Sheep Overexpressing Ovine β-Catenin

**DOI:** 10.3390/ijms20030620

**Published:** 2019-01-31

**Authors:** Jiankui Wang, Kai Cui, Zu Yang, Tun Li, Guoying Hua, Deping Han, Yanzhu Yao, Jianfei Chen, Xiaotian Deng, Xue Yang, Xuemei Deng

**Affiliations:** Key Laboratory of Animal Genetics, Breeding, and Reproduction of the Ministry of Agriculture & Beijing Key Laboratory of Animal Genetic Improvement, China Agricultural University, Beijing 100193, China; jiankui4955@163.com (J.W.); fly87217@126.com (K.C.); yangzu-198629@163.com (Z.Y.); 19925489460@163.com (T.L.); huaguoyingnb@163.com (G.H.); handeping1984@163.com (D.H.); yzyao2011@163.com (Y.Y.); 15664451908@163.com (J.C.); dxt9509@cau.edu.cn (X.D.); snow_yang2017@163.com (X.Y.)

**Keywords:** sheep, β-catenin, wool follicle, transgene, transcriptome

## Abstract

β-Catenin is an evolutionarily conserved molecule in the canonical Wnt signaling pathway, which controls decisive steps in embryogenesis and functions as a crucial effector in the development of hair follicles. However, the molecular mechanisms underlying wool production have not been fully elucidated. In this study, we investigated the effects of ovine β-catenin on wool follicles of transgenic sheep produced by pronuclear microinjection with a skin-specific promoter of human keratin14 (k14). Both polymerase chain reaction and Southern blot analysis showed that the sheep carried the ovine β-catenin gene and that the β-catenin gene could be stably inherited. To study the molecular responses to high expression of β-catenin, high-throughput RNA-seq technology was employed using three transgenic sheep and their wild-type siblings. These findings suggest that β-catenin normally plays an important role in wool follicle development by activating the downstream genes of the Wnt pathway and enhancing the expression of keratin protein genes and keratin-associated protein genes.

## 1. Introduction

The skin provides a barrier against environmental assault from foreign substances and organisms. The outer skin of mammals is covered by multilayered epithelial cells, including hair follicles and glandular structures [1]. Although the upper part of the hair follicle and the dermal papilla are formed during embryonic development, these structures are fixed in postnatal life [2]. The papilla is a large structure at the base of the hair follicle [3], and a series of cell signaling processes transfer signals from one cell to another between the epithelium and dermis during embryonic development and hair follicle morphogenesis [4]. In postnatal animals, the lower portion of the hair follicle is responsible for hair follicle cycles of anagen, catagen, and telogen, and molecular signatures in the dermal papilla are thought to induce the proliferation of stem-like cells in the epithelium of the hair follicle bulge and the migration of daughter cells to the lower portion of the hair follicle [2,5]. Previous research has shown that several key signals, such as fibroblast growth factor (FGF), bone morphogenic protein (BMP), and transforming growth factor β2 (TGF-β2) have been implicated in hair follicle induction [6]. Recent studies have shown that WNT signaling pathways, particularly the essential molecule β-catenin, play important roles in hair follicle formation and periodic growth [7].

As a multifunctional molecule, β-catenin plays various roles in WNT signaling and interacts with a number of proteins, including the actin-binding protein facin and presenilins. Moreover, β-catenin also has a role in Ca^2+^-dependent cell adhesion [8]. In the canonical WNT pathway, β-catenin contributes to the transduction of the signal to the nucleus and associates with transcription factors from the T-cell factor/lymphoid enhancer factor family to drive the transcription of WNT/β-catenin target genes. The mRNA expression level of β-catenin is increased during hair follicle formation in embryonic development. In contrast, blocking the expression of β-catenin disrupts the formation of normal hair follicles [9]. In vitro studies have indicated that two particular phenotypes are acquired by knockout of β-catenin: (1) placode formation is not observed during embryo development; and (2) hair loss appears in the first hair cycle [7]. Compared with wild-type mice, overexpression of a truncated β-catenin in the skin of mice was found to cause a rapid increase in the number and size of hair follicles, and the hair follicles in telogen could enter into the anagen in advance [10]. In the “bulge activation hypothesis”, when signals from dermal papilla transiently activate the quiescent hair follicle bulge stem cells to proliferate, the hair follicle turns into a new anagen phase; β-catenin plays an essential role in this process [3].

Aohan fine-wool sheep are a wool–meat breed generated from crossbreeding with Inner Mongolian local sheep and the former Soviet Union merino sheep [11]. These sheep are known for their slaughter rate, crude feed tolerance, and wool fineness. In recent years, the demand for wool has increased rapidly with the development of the wool textile industry. However, wool production cannot meet this increasing demand, and novel approaches for improving wool production are needed. As the origin of wool fibers, hair follicles have direct effects on the production and quality of ovine cashmere.

In the present study, we examined the role of the *K14* gene promoter in directing the overexpression of β-catenin in the wool follicle in transgenic sheep and the effects of β-catenin overexpression on wool production in transgenic sheep.

## 2. Results

### 2.1. Transgenic Sheep

To investigate the functions of β-catenin in wool follicles, the pK14-β-catenin-enhanced green fluorescent protein (EGFP)-N1 plasmid was successfully constructed (Figure 1A). As shown in Figure 1B, the transgene fragments were generated by Ase1 and Age1 double enzyme digestion and then microinjected into Aohan fine-wool sheep zygotic pronuclei. In total, 155 fertilized oocytes were microinjected with a linearized transgene construct and transferred to 68 recipients; 62 lambs were produced, of which nine were transgenic as determined by PCR (Figure 1C) and Southern blot analysis (Figure 1E). One ram (B025) was used for artificial propagation by superovulation artificial insemination and embryo transfer. Twenty-six lambs were produced, six of which were transgenic as identified by PCR (Figure 1D) Southern blot analysis of DNA from the transgenic sheep further indicated the existence of foreign DNA in the transgenic sheep (Figure 1F). The transgenic sheep were healthy and showed no defects.

### 2.2. Characterization of Transgene Expression in Transgenic Sheep and Their Wild-Type Siblings

To compare transgene expression level, relative transgene mRNA levels were measured by quantitative reverse transcription polymerase chain reaction (qRT-PCR). The results indicate that relative transgene expression levels in the skin of F0 transgenic sheep were significantly higher than those in their wild-type siblings (*P* < 0.01; Figure 2A,B). We investigated the ovine β-catenin expression in F1 transgenic sheep skin tissues from line B025 for further study. The results show that transgene expression in the skin was significantly higher than that in their full siblings (*P* < 0.01; Figure 2E,F). Additionally, Western blotting results show that β-catenin protein in F0 (Figure 2C) and F1 (Figure 2G) was expressed at higher levels in transgenic sheep skin than in their wild-type siblings. ImageJ was used to analyze the protein expression level of F0 (Figure 2D) and F1 (Figure 2H).

Immunohistochemical (IHC) assays were performed to determine the localization of β-catenin in the wool follicles (Figure 3). The IHC analysis showed that β-catenin protein in transgenic sheep skin was found in the inner root sheath and outer root sheath (Figure 3A). However, β-catenin was only expressed in the outer root sheath of wild-type siblings (Figure 3C). In addition, β-catenin protein was expressed in Dermal papilla of transgenic sheep (Figure 3B) compare with the Dermal sheath of wild-type sibling (Figure 3D).

### 2.3. Advantage of Overexpressing Ovine β-Catenin in Wool Production

Clean fleece weight was obtained at approximately 14 months of age, and clean wool production on average was 26.49% greater (*P* = 0.0051) in the transgenic sheep than in their nontransgenic full siblings (Figure 4A) by Valency T tests. However, body weight did not differ significantly (*P* = 0.3354) between the transgenic sheep and their nontransgenic full siblings (Figure 4B). Table 1 gives the details of each sample.

### 2.4. Analysis of Wool Follicle Density of Transgenic Sheep

The frozen sections of skin showed different wool follicle growth patterns and wool follicle densities between transgenic sheep and their nontransgenic full siblings. The wool follicles of transgenic sheep showed a more uniform distribution than that in nontransgenic sheep (Figure 5A–J). This difference was also observed on longitudinal sections of skin (Figure 6A–J). This wool follicle phenotype characteristic led to the higher wool follicle density in transgenic sheep than in nontransgenic full siblings by wool follicle density analysis (Figure 5K,L). There were no significant difference between transgenic sheep and nontransgenic full siblings in the wool length (Appendix A) and diameter (Appendix A).

### 2.5. Identification of Expressed Transcripts in the Sheep Skin Transcriptome

In this study, 58,356,618–63,795,138 raw reads were generated for each sample (Appendix A). We obtained 20,532 transcripts from the groups after quality control (Appendix A). About 85% of the total reads were mapped to the sheep reference genome, and about 75% of the total reads were mapped to the reference genome only at one site (Appendix A). Information for read density on chromosomes is provided in the Appendix A. The top 10 annotated transcripts, ranging from 3509 to 7628 fragments per kilobase of exons per million fragments mapped (FPKM) reads (Figure 7), were ranked by abundance and included the keratin protein genes or keratin-associated protein genes *LOC101112404* (keratin-associated protein 3-2), *LOC101104027* (keratin-associated protein 7-1-like), *LOC101103772* (keratin-associated protein 11-1), *KRTAP6-1*, *KRTAP1-1*, *KRT25*, and *LOC101106046* (keratin-associated protein 13-1-like).

### 2.6. Functional Analysis of Differentially Expressed Genes (DEGs)

To elucidate the biological processes in transgenic sheep, gene ontology (GO) assignments were performed. Based on sequence homology, 76 identified upregulated DEGs and 37 downregulated genes were categorized into major functional groups, including biological processes (BPs), cellular components (CCs), and molecular functions (MFs; Figure 8). Among these terms, the first functional categories of BPs, CCs, and MFs were related to hair follicle development (Figure 7). Of these upregulated genes, *FN1* and matrix metalloproteinase-7 (*MMP-7*) were found to be direct target genes of β-catenin [3].

### 2.7. Identification of DEGs between the F0 Transgenic Sheep and Wild-Type Siblings

Approximately 113 DEGs between the two groups were identified by paired t-tests. By applying the criteria of ≥2-fold change and *P* < 0.05 as cut off values, 76 upregulated and 37 downregulated DEGs were screened out. Further annotation showed that 11 keratin protein genes or keratin-associated protein genes were upregulated in the transgenic sheep, whereas none of these keratin and keratin-associated protein genes were found in downregulated genes in transgenic sheep. In addition, β-catenin and its target genes (*MMP-7* and *FN1*) were significantly upregulated in transgenic sheep compared with that in their wild-type siblings (Table 2).

### 2.8. Validation of DEGs by Real-Time PCR

Eight DEGs were selected for validating the DEGs identified from RNA-seq data, including four upregulated genes (β-catenin, *KRT25*, *KRT71*, and *KRT79*) and four downregulated genes (*MPC1*, *KRTDAP*, *ASAP2*, and *ASB7*). The expression patterns of DEGs between these two groups were confirmed by the results of RT-PCR (Figure 9A). The correlation of fold change between RNA-seq and qRT-PCR was analyzed, and the correlation coefficient was 0.9257 (Figure 9B).

## 3. Discussion

Hair follicle morphogenesis is a fine example of the exchange of information between epithelium and mesenchyme, which is the basis of organogenesis [4]. The molecular mechanisms of these interactions are still largely unclear, but are likely to involve several signaling molecules, including WNTs [20], FGFs [21], ectodysplasin A receptors [22,23], BMPs [24], Sonic hedgehog proteins [25], TGF-βs [26], and insulin-like growth factors [27]. Among these mechanisms, the Wnt pathway plays an essential role in many aspects of development and tumorigenesis [28]. Nuclear localization of β-catenin is downstream of the Wnt pathway, which is localized in the cytoplasm and membrane [29]. A conditional mutation in β-catenin in the epidermis and hair follicles demonstrated that the formation of placodes, which generate hair follicles, was blocked [7], suggesting that activation of the canonical Wnt signaling pathway was necessary for hair follicle formation.

To better understand the effects of β-catenin on wool development, stable and heritable skin-specific β-catenin transgenic sheep were obtained for the first time. In our study, the wool follicle density was significantly greater in transgenic sheep than in their wild-type siblings of the same sex. Consistent with our study, an overall increase of 300% ± 30% in the hair germ density was found in β-catenin transgenic mice skin relative to that in control skin [10].

Furthermore, we found that the wool follicles in transgenic sheep skin were distributed more homogeneously than their wild-type siblings by skin observation on cross-sections; this was an abnormal wool follicle growth characteristic because the wool follicles were accustomed to growth in clumps, called wool follicle groups [30]. The overexpression of β-catenin was thought to induce new wool follicle outgrowth and promote the development of wool follicles such that the classical development characteristics (wool follicle groups) of wool follicles would be disrupted. The same conclusion was reached by analysis of longitudinal sections of skin between transgenic sheep and their wild-type siblings, which was similar to a study in which activation of the β-catenin signal in the adult mouse epidermis was sufficient to induce new hair follicles [31] and ectopic hair outgrowth [32].

To analyze differences in the activity of Wnt signaling between transgenic sheep and negative control sheep, the localization of β-catenin was observed by IHC staining. The results show that β-catenin mRNA and protein were significantly upregulated in transgenic sheep skin compared with those in wild-type siblings of the same sex. Interestingly, our work suggested more active expression of β-catenin protein in the dermal papilla of transgenic wool follicles compared with that in nontransgenic full siblings according to the results of IHC staining. Furthermore, IHC staining results show that β-catenin protein was not only expressed in the outer root sheath but also in the inner root sheath (IRS) of transgenic sheep wool follicles. In contrast, expression was observed only in the outer root sheath of nontransgenic full siblings. At present, we have not found any evidence to explain the possible reasons this differential expression pattern occurred; however, KRT5, KRT25, and KRT71 were found to be expressed in the Henle layer, Huxley layer, and IRS cuticle [33]. In addition, these three keratin protein genes were mainly expressed in transgenic sheep compared with their nontransgenic full siblings. These findings suggested that there may be a connection between β-catenin and keratin protein genes and a link between hair shaft shape as natural traits and keratin protein genes [34]. In addition, previous studies have shown that the dermal papilla is of prime importance in hair follicle morphogenesis. Papillae can induce hair growth after implantation into follicles [35,36] and can interact with the skin epidermis to form new hair follicles [35]. Accordingly, we suspect that β-catenin can promote the development of dermal papilla and IRS in transgenic sheep wool follicles, allowing the growth state to be more active than that in wool follicles of wild-type sibling sheep.

There is a complex interactive process between molecular signaling transduction and hair morphogenesis. The WNT pathway plays an essential role in the formation of hair placodes during embryogenesis [37], and β-catenin is required for the differentiation of skin stem cells in the adult, making it the key molecule in the Wnt pathway. In the absence of β-catenin, these stem cells are unable to adopt the fate of hair keratinocytes and instead differentiate into epidermal keratinocytes [7]. Additionally, inactivation of the β-catenin gene in the dermal papilla of anagen hair follicles gives rise to obviously reduced proliferation of progenitors (dermal sheath), which produce the hair shaft [38]. However, few studies have examined sheep wool follicle development and changes in the molecular network in transgenic sheep overexpressing β-catenin. In our study, high-throughput sequencing was used to obtain DEGs from three pairs of full sibling sheep of the same sex. Transcripts encoding 113 DEGs were identified, with 72 significantly upregulated genes in transgenic sheep relative to those in nontransgenic sheep, including β-catenin and its direct or indirect target genes (*MMP-7*, *FN1*, *KRT79*, *KRT2.11*, *KRT8*, *KRT71*, *KRT5*, *KRT25*, and *KRTAP1-1*). This result show that the WNT pathway was more active in transgenic sheep than in nontransgenic sheep and that DEGs, associated with hair follicle development were mainly upregulated in transgenic sheep relative to those in nontransgenic sheep. In addition, GO analysis was used to identify functional categories of those DEGs obtained from the global analysis for in-depth study. For the upregulated genes in transgenic sheep, analysis of the top ranking GO terms revealed an enrichment in genes related to “hair follicle morphogenesis” and “keratin filament”, which was further evidence that the β-catenin skin-specific transgenic sheep model was successfully constructed.

## 4. Materials and Methods

### 4.1. Animals and Treatments

Three pairs of full siblings (one pair of males and two pairs of females), at 12 months of age, were selected and divided into transgenic (+/−) and nontransgenic (WT) groups. Skin tissue was obtained from the scapular region of each sheep and frozen in liquid nitrogen directly. All of the procedures and experiments performed in this study were approved by the Animal Care and Use Committee of China Agricultural University (Approval no. XK257).

### 4.2. K14-β-Catenin-EGFP Plasmid Construction

The pEGFP-N1 vector was digested with restriction enzymes AseI and AfeI to remove the CMV promoter and then inserted into the human *K14* promoter (~2.0 kb) between the AseI and AfeI sites to generate the skin-specific expression plasmid K14-EGFP. Subsequently, the K14-EGFP plasmid was digested with restriction enzymes SacII and BamHI, and the complete coding sequence of the ovine β-catenin gene was inserted between the SacII and BamHI sites. The resulting K14-β-catenin-EGFP plasmid was then verified by sequencing with the primers shown in Table 1 (SinoGenoMax Co. Ltd., Beijing, China).

### 4.3. Generation of Transgenic Sheep

All of the procedures and experiments performed in this study were approved by the Animal Care and Use Committee of China Agricultural University (Approval no. XK257). Linear DNA fragments were obtained by double digestion of the K14-β-catenin-IRES-EGFP plasmid with AseI/Age1 and re-suspended in sterile ddH_2_O at a concentration of 20 ng/μL. The linearized construct was injected into the pronuclei of the fertilized eggs flushed from donor sheep. The K14-β-catenin-IRES-EGFP transgene-positive founder ram was mated with the wild-type to obtain F1 generation sheep using artificial insemination, superovulation, and embryo transfer.

### 4.4. Positive Identification of F0 and F1 Transgenic Sheep

Genomic DNA extracted from skin tissues of transgenic sheep using the TIANamp Genomic DNA Kit (TIANGEN, Beijing, China) was used for PCR and Southern blotting analysis. PCR analysis was performed using specific primers (forward primer: 5′-CTGCCTGGATTTCTCTTTGA-3′; reverse primer: 5′-CCATTGTCCACGCAAGATTT-3′). The 875-bp PCR product contained the partial human K14 promoter sequence and partial ovine β-catenin sequence. PCR amplification was carried out as follows: denaturation for 5 min at 95 °C; then 35 cycles of 30 s for 95 °C, 30 s for 58 °C, and 72 °C for 1 min; 72 °C for 10 min; and hold at 12 °C. The positive sheep identified by PCR were subjected to Southern blot analysis.

For Southern blot analysis, 25 μg genomic DNA obtained from skin tissues was digested with the restriction enzyme BstEII at 60 °C for 20 h. A 739-bp probe, amplified by F6 (CAAGAAAGCCCAAAACAC) and R6 (TAGCGTCTCAGGGAACAT) from the promoter and β-catenin sequences, was labeled with a PCR digoxigenin probe synthesis kit (Roche, Basel, Switzerland). Hybridization and washing were performed using a DIG-High Prime DNA Labeling and Detection Starter Kit II (Roche) according to the manufacturer’s instructions.

### 4.5. Analysis of Transgene Expression of F0 and F1 Sheep

Total RNA was isolated from the skin of transgenic sheep using an RNAprepPure Tissue Kit (TIANGEN), according to the manufacturer’s instructions. The first-strand cDNA was synthesized from 1–2 μg of purified total RNA using an ImProm-II Reverse Transcription System (Promega, Beijing, China). PCR was performed using the primers shown in Table 1. Tissue-specific expression analysis of β-catenin was performed using qRT-PCR with an iQ5 real-time PCR detection system (Bio-Rad, Hercules, CA, USA) relative to the expression of glyceraldehyde-3-phosphate dehydrogenase (GAPDH), which was used as a housekeeping gene (internal reference). The relative β-catenin expression levels were assessed in triplicate and calculated using the 2^−ΔΔ*C*t^ method.

### 4.6. Protein Analysis of F0 and F1 Sheep

For Western blot analysis, protein from F0 and F1 sheep skin tissues was extracted using RIPA buffer (BeYoTime, Suzhou, China) according to the manufacturer’s instructions. A BCA Protein Assay Kit from BeYoTime was used to determine the protein concentrations, and bovine serum albumin was used as a standard. Equal amounts of total cellular protein (20 μg/lane) were resolved on sodium dodecyl sulfate-polyacrylamide gels and transferred onto polyvinyl-difluoride membranes. The membranes were hybridized with 1:1000 dilutions of anti-β-catenin antibodies (cat no. E247; Abcam, Cambridge, UK) or antiβ-actin antibodies (cat. no. AA128; BeYoTime) and incubated at 37 °C for 1 h after blocking with blocking buffer for 1 h at room temperature. After washing five times with Tris-buffered saline with Tween20 (TBST) for 5 min, membranes were incubated with horseradish peroxidase-conjugated secondary antibody (1:10,000) for 1 h at 37 °C, washed six times with TBST for 5 min each, and visualized using an enhanced chemiluminescence system (cat. no. P0018S; BeYoTime).

### 4.7. Immunohistochemistry

Frozen sections of skin were washed with distilled water three times, incubated in 3% hydrogen peroxide solution for 20 min at room temperature, washed with distilled water again, and blocked with blocking buffer for 30 min at room temperature. The sections were incubated with primary mouse anti-human β-catenin antibodies (Abmart, Shanghai, China; 1:200) overnight at 4 °C. After rinsing with phosphate-buffered saline (PBS), the sections were treated with goat anti-mouse immunoglobulin G conjugated to horseradish peroxidase (Zhongshan GoldenBridge Biotechnology, Beijing, China) for 1 h at room temperature in the dark. After washing with PBS three times, the sections were incubated with 3, 3′-diaminobenzidine (DAB) for 10 min at room temperature in the dark. After DAB staining, slides were rinsed three times in deionized water for 30 s and then stained with hematoxylin. After treatment with hydrochloric acid, ethyl alcohol, and dimethylbenzene, the sections were embedded in Permount Mounting Medium and observed under a microscope.

### 4.8. Clean Fleece Weight

At yearling shearing, body weight was recorded, the fleece was weighed, and a sample was taken for scouring. Clean dry fleece weight and yield were then assessed. To measure clean dry fleece weight, greasy wool samples were kept in a conditioned environment (20 °C, 65% relative humidity (RH)) for 24 h, washed in a sample wool scourer with nonionic detergent at 65 °C, dried at 60 °C for 20 min, maintained in a conditioned environment at 20 °C and 65% RH for 24 h, and then weighed. Yield was calculated as clean weight divided by greasy weight and expressed as a percentage.

### 4.9. RNA Extraction and Transcriptome Sequencing

RNA was extracted from skin using TRIzol Reagent (Invitrogen, San Diego, CA, USA), following the manufacturer’s protocol. Next, 1% agarose gels were used to determine the quality of the RNA. RNA concentration was measured using a Qubit RNA Assay Kit in a Qubit 2.0 Fluorometer (Life Technologies, CA, USA). A NanoPhotometer (IMPLEN, Munchen, Germany) and RNA Nano 6000 Assay kit for a Bioanalyzer 2100 system (Agilent Technologies, Foster city, CA, USA) were used to check the RNA purity, and RNA samples passing the quality tests were used for further analyses.

Three micrograms of RNA were used for library construction. We use an IlluminaTruseq RNA sample preparation Kit (Illumina, San Diego, CA, USA) to produce sequencing libraries. Poly-T oligo-attached magnetic beads were used to purify mRNA from total RNA. The mRNAs were broken into short fragments, and random oligonucleotides and Superscript II were used to generate the first-strand cDNA. Second-strand cDNA was synthesized using DNA polymerase I and RNase H. In addition, sequencing adaptors were used to purify and modify double-stranded cDNAs, and a cDNA library was created using suitable fragments, which were selected by gel purification and enriched by PCR amplification. Subsequently, the library was used for sequencing on an IlluminaHiseq 2000 platform (Illumina, San Diego, CA, USA).

### 4.10. Sequence Reads Mapping, Assembly, and Annotation

Raw reads in the fastq format were first processed through in-house Perl scripts. During this step, clean reads were obtained by removing reads containing adapters, reads containing poly-N sequences, and low-quality reads from raw data. At the same time, Q30, GC-content, and sequence duplication levels of the clean data were calculated. All high-quality clean data were used for subsequent analyses.

The sheep reference genome and gene model annotation files were downloaded from the genome website (http://www.sheephapmap.org/news/OARv2p0.php). Bowtie v0.12.8 was used to build an index of the reference genome [39], and TopHat v1.4.0 was used when paired-end clean reads were aligned to the reference genome [40]. Clean reads were aligned to the reference genome using SOAP2 [41]. To eliminate the PCR interference and ambiguous mapping, duplicated reads and multimapped reads were filtered from the alignment results.

### 4.11. Quantification and Analysis of DEGs

The gene expression levels were estimated by HTSeqv0.5.3 (http://www-huber.embl.de/users/anders/HTSeq) for each sample. The FPKM was obtained according to the mapped transcript fragments, transcript length, and sequencing depth. Differences in gene expression between control and case groups were analyzed based on the DESeq R package (1.10.1) [42]. To control the false discovery rate, the Benjamini and Hochberg’s approach were used to adjust the *P* values [43]. Genes with an adjusted *P* value of less than 0.05 found by DESeq were regarded as differentially expressed.

### 4.12. GO and Kyoto Encyclopedia of Genes and Genomes (KEGG) Enrichment Analysis of DEGs

GO enrichment analysis of differentially expressed transcripts was performed using the GOseq R package [44], such that the gene length bias could be adjusted. To discover and cluster the biological functions of DEGs, the KEGG pathways were analyzed by Database for Annotation, Visualization and Integrated Discovery (DAVID) v6.8 (https://david.ncifcrf.gov/). DAVID provides a comprehensive set of functional annotation tools for investigators to understand biological meaning behind large lists of genes. Functional groups with at least two DEGs in the background terms were selected, and those with a *P* value of less than 0.05 were considered significantly overexpressed. The top 10 functional groups, such as MF, CC, and BP, were selected to show in the picture. KOBAS [44] software was used to test the statistical enrichment of DEGs in KEGG pathways.

### 4.13. Real-Time PCR Validation of DEGs

DEGs selected from RNA-seq data were validated using qPCR, which was performed with SYBR Premix ExTaq (TaKaRa, Dalian, China) on an ABI 7500 Real-Time PCR System (Applied Biosystems, Foster City, CA, USA). cDNA synthesis was performed using Quantscript RT Kit Quant cDNA (Tiangen Biotech, Beijing, China) with approximately 400 ng of total RNA as the template. SYBR Premix ExTaq (Takara, Dalian, China) was used for real-time PCR analysis on an ABI7500 Real-Time PCR System (Applied Biosystems) with cycling conditions of 95 °C held for 30 s; 42 cycles at 95 °C for 5 s and 59 °C for 20 s; followed by the programmed dissociation analysis from 95 °C to 60 °C to verify the amplification authenticity. The β-actin gene was used as a reference control. Each plate was repeated three times in independent runs for all references. Eight genes were chosen (β-catenin, *KRT25*, *KRT71*, *KRT79*, *MPC1*, *KRTDAP*, *ASAP2*, and *ASB7*) for the validation. PCR was performed using the primers shown in Table 3, all of which were self-designed except for KRT71 [45]. Gene expression was evaluated by the 2^−ΔΔ*C*t^ method [46].

## 5. Conclusions

In our study, heritable, stable expression of ovine β-catenin in the wool skin of transgenic sheep was obtained. New wool follicles were induced by high expression of β-catenin in sheep. The wool follicle density and fleece weight were increased. These sheep were expected to be an excellent resource for breeding of sheep producing high-quality wool. The results of high-throughput RNA-seq showed that β-catenin and some of its direct or indirect target genes were upregulated in transgenic sheep, demonstrating the high activity of the Wnt pathway in these transgenic sheep. Thus, our findings established a new animal model for exploring the molecular mechanism of wool follicle development and for additional analysis of the Wnt pathway. Further studies are still needed to elucidate the mechanisms of β-catenin gene expression in the inner root and outer root of the sheath of transgenic sheep; such studies may reveal new approaches for regulating hair follicle development by the Wnt pathway.

## Figures and Tables

**Figure 1 ijms-20-00620-f001:**
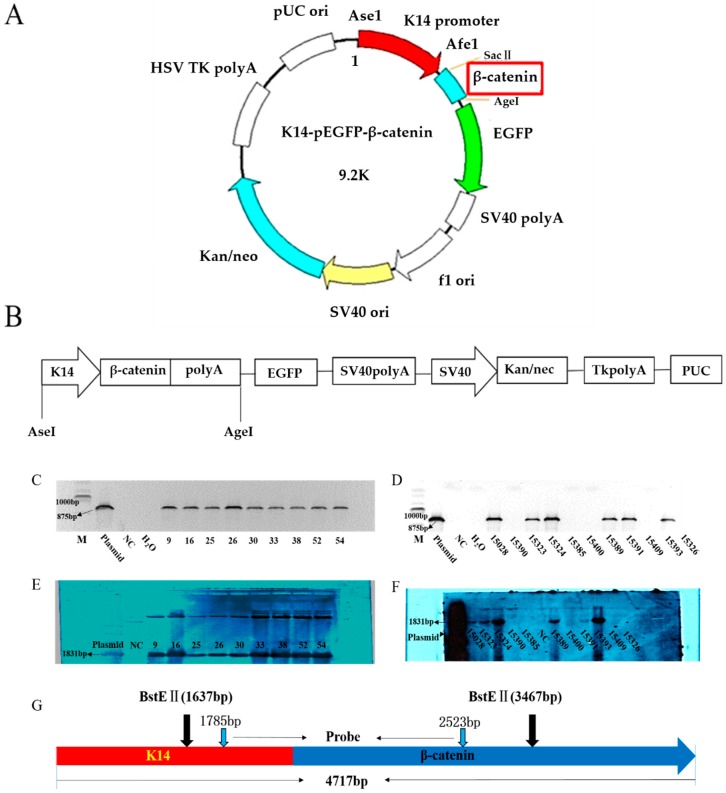
Generation of ovine β-catenin transgenic sheep: (**A**) the recombinant plasmid structure of K14-β-catenin-N1-EGFP; (**B**) the stick diagram of K14-β-catenin-N1-EGFP; (**C**,**D**) the PCR identification of transgenic sheep; (**E**) Southern blot of DNA from F0 transgenic sheep skin (NC: sex- and age-matched wild-type control; plasmid: positive control); (**F**) Southern blot of DNA from F1 transgenic sheep skin; and (**G**) schematic diagram of restriction endonuclease digestion by Southern blotting.

**Figure 2 ijms-20-00620-f002:**
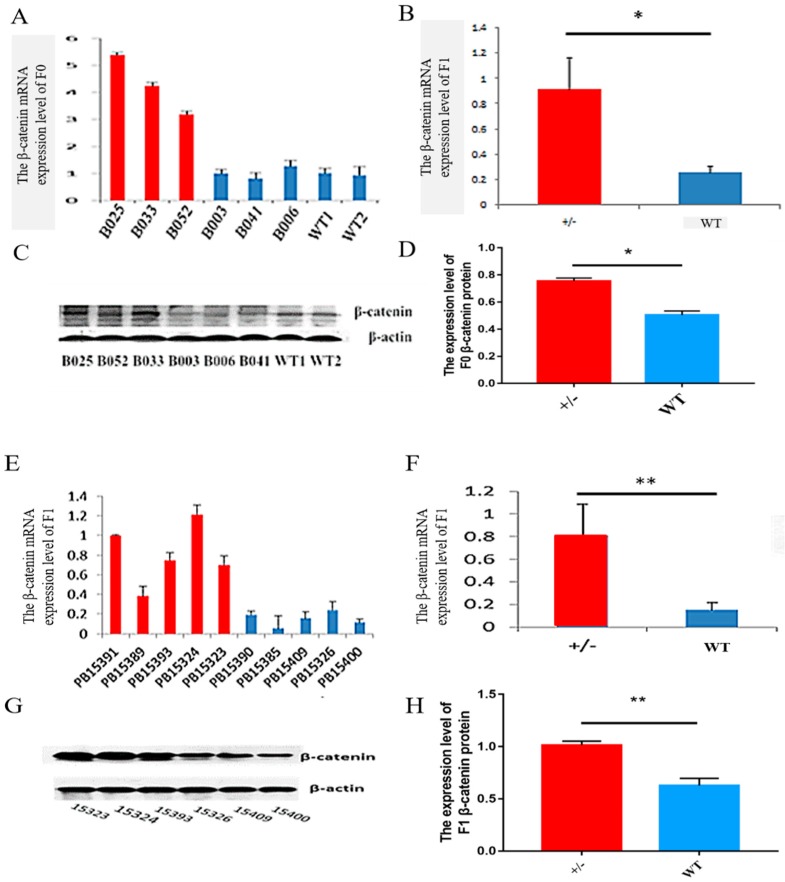
The expression of β-catenin in the skin of F0 and F1 sheep. The relative quantities of β-catenin mRNA in F0 (**A**,**B**) and F1 (**E**,**F**) were detected via qRT-PCR using the 2^−ΔΔ*C*T^ method, with *GAPDH* as the internal control. Each experimental group contained at least three replicates, and qRT-PCR was performed in triplicate for each sample. Bars with common lowercase letters are not significantly different at the level of 5%. WT1 and WT2 are wild-type control sheep. The analysis of β-catenin protein expression in F0 (**C**,**D**) and F1 (**G**,**H**) was performed using Western blot analysis at 12 months with β-actin as the internal control. The bands for β-catenin and β-actin proteins were quantified with ImageJ (http://rsb.info.nih.gov/ij). F0 transgenic sheep: B025, B033, and B052; F0 wild-type siblings: B003, B006, and B041; wild-type sheep: WT1 and WT2; F1 transgenic sheep: PB15391, PB15389, PB15393, PB15324, and PB15323; and F1 wild-type siblings: PB15390, PB15385, PB5409, PB15326, and PB15400. The significance of differences in β-catenin mRNA and protein expression levels between transgenic sheep and wild-type siblings was analyzed by paired Student’s t tests. * *P* < 0.05; ** *P* < 0.01 for comparisons between the two groups.

**Figure 3 ijms-20-00620-f003:**
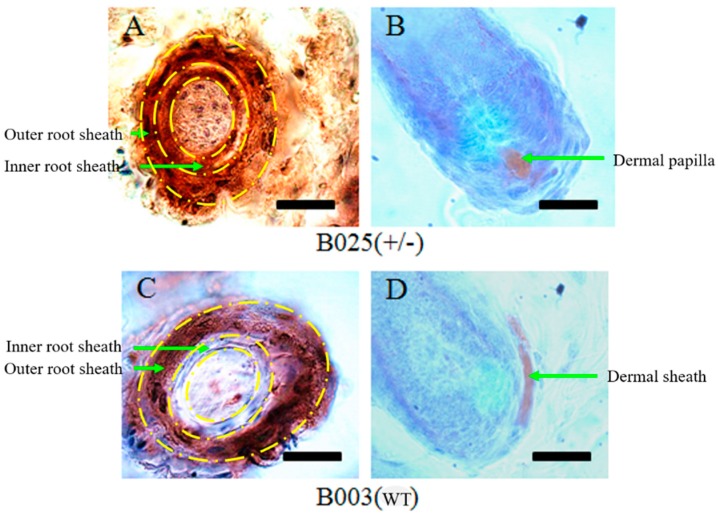
Localization of β-catenin protein in the wool follicle: (**A**,**B**) the expression of β-catenin in the inner and outer root sheath and hair dermal papilla in the transgenic sheep; and (**C**,**D**) the expression of β-catenin in the outer root sheath of the wool follicle in the wild-type sibling control. The brown granules show the localization of β-catenin in the wool follicle. Scale bars: 10 μm.

**Figure 4 ijms-20-00620-f004:**
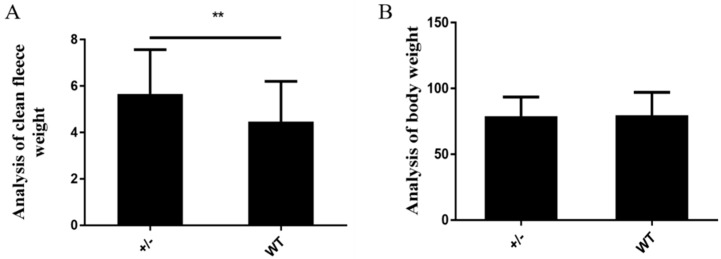
Comparison of clean fleece weights (**A**) and body weights (**B**) between the transgenic sheep and their wild-type siblings. The significance of differences in body weights and clean wool weights between transgenic sheep and wild-type siblings was analyzed by paired t-tests. ** *P* < 0.01 for comparisons between the two groups.

**Figure 5 ijms-20-00620-f005:**
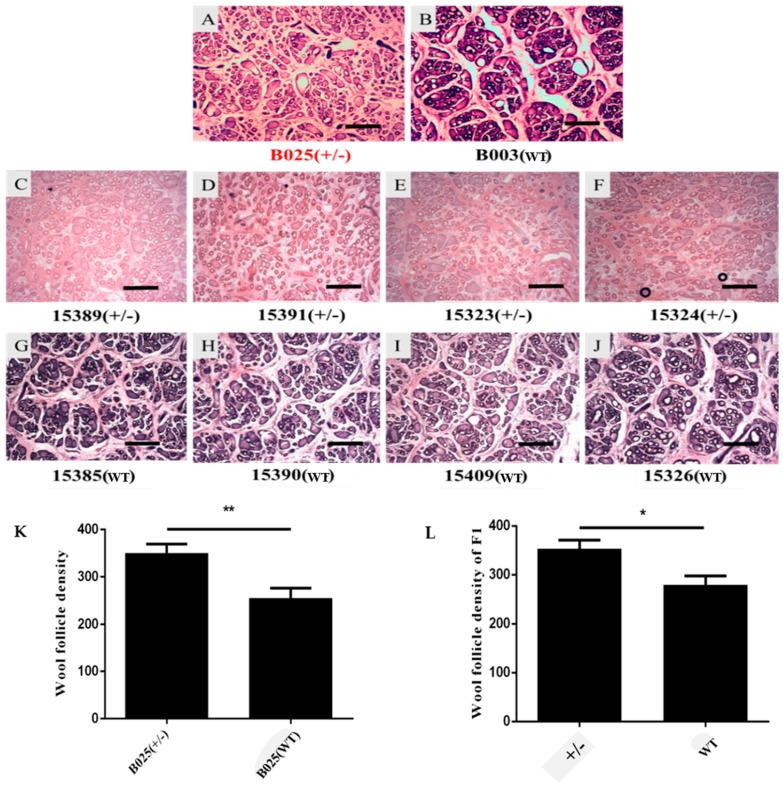
Analysis of wool follicle density in F0 and F1: (**A**,**B**) Hematoxylin–eosin staining showing the hair follicle morphological characteristics of a transgenic sheep (B025) and its wild-type sibling (B003); (**C**–**F**) transgenic sheep of F1; (**G**–**J**) wild-type siblings longitudinally corresponding to the transgenic sheep of F1; (**K**) wool follicle density comparison of the F0 pair; and (**L**) wool follicle density comparison of the F1 pairs. * *P* < 0.05, ** *P* < 0.01. Scale bars: 100 μm.

**Figure 6 ijms-20-00620-f006:**
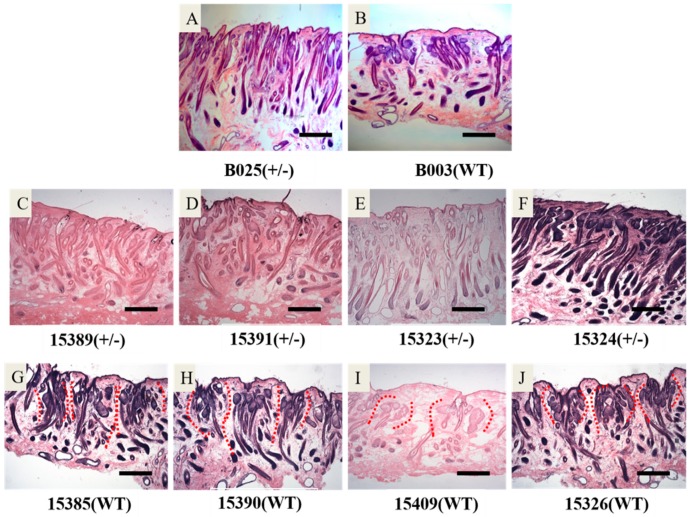
Developmental characteristics of hair follicles between transgenic sheep and their nontransgenic full siblings: (**A**) K14-β-catenin transgenic ram (B025); (**B**) the nontransgenic full sibling of B025, B003; (**C**–**J**) slices from the offspring of the ram B025 generated by synchronous estrus, superovulation, and artificial insemination; (**C**–**F**) slices from the F1 transgenic sheep; and (**G**–**J**) slices from the wild-type siblings longitudinally corresponding to the transgenic sheep of F1.

**Figure 7 ijms-20-00620-f007:**
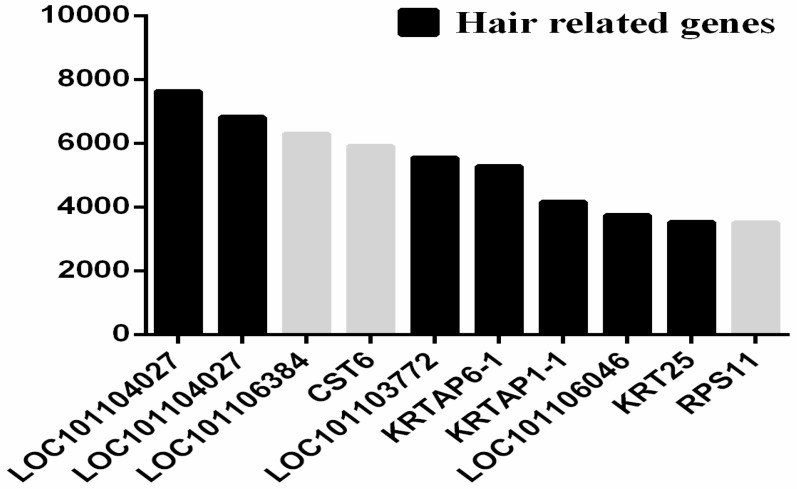
Expression of the top 10 most highly expressed genes in sheep skin. The *x*-axis shows gene ID, and the *y*-axis shows gene expression level (FPKM).

**Figure 8 ijms-20-00620-f008:**
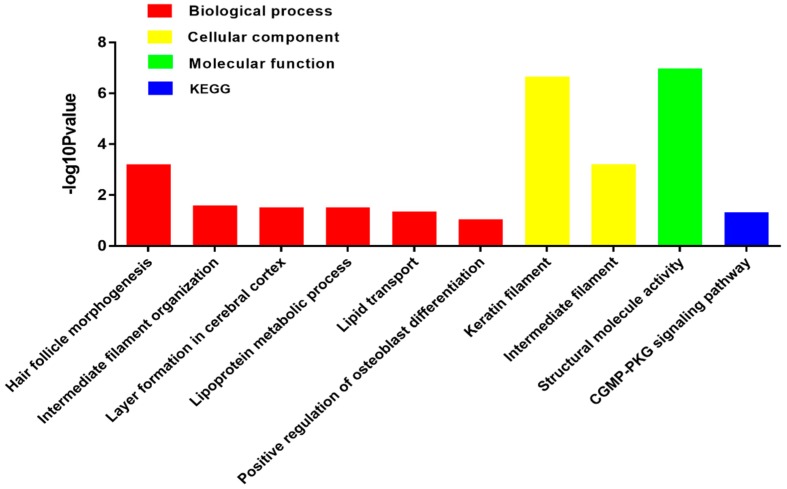
Functional categorization of DEGs based on known genes in the Uniprot database. The *x*-axis shows GO and KEGG terms; the *y*-axis shows the negative log10Pvalue.

**Figure 9 ijms-20-00620-f009:**
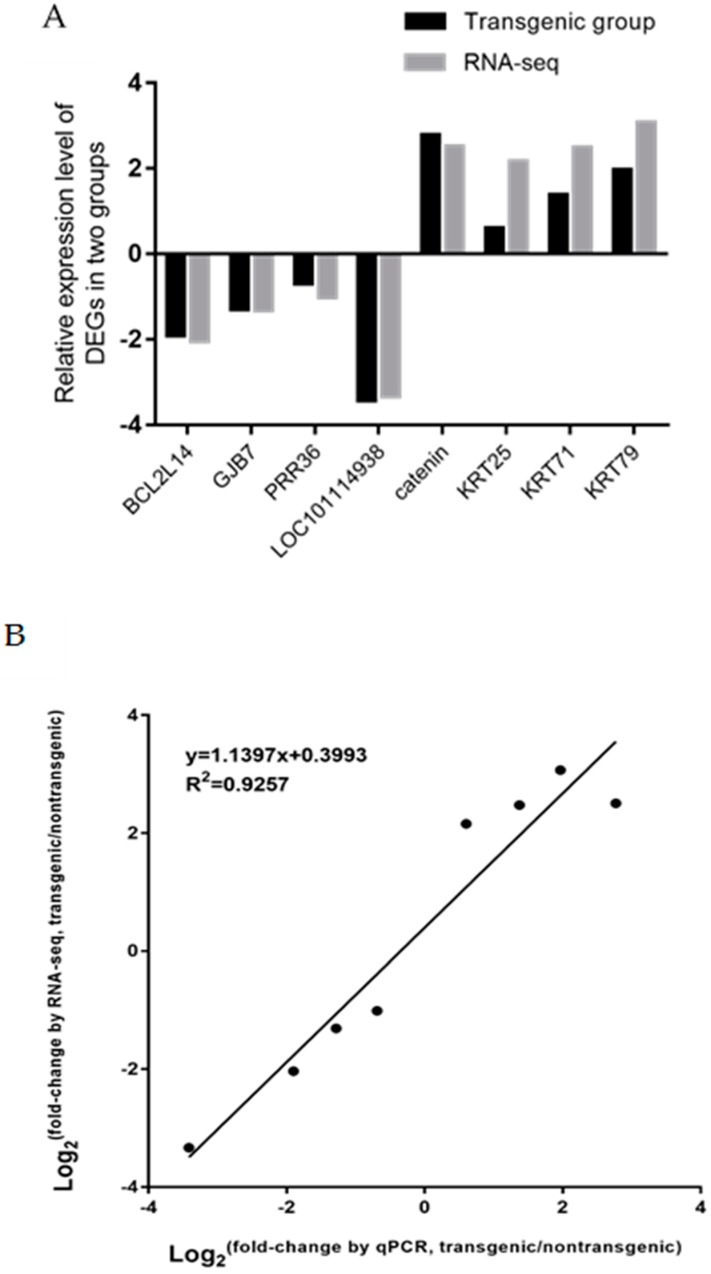
The expression levels of DEGS validated by qPCR (**A**); and comparison between RNA-seq and qPCR by correlational analyses (**B**).

**Table 1 ijms-20-00620-t001:** Analysis of wool production and body weight in F1.

Ear Tag	Donor	Donor	Sex	Body	Yearling	Clean	Clean	GMO (+/−)
	rams	ewes		Weight (kg)	Shearing (kg)	Fleece Yield (%)	Fleece Weight (kg)	Non-GMO (WT)
15323	B025	ZA94822	♂	85.35	10.85	50.14	5.44019	+/−
15409			♂	88.5	8.4	58.63	4.92492	WT
15324	B025	ZA94822	♀	54.6	4.45	63.48	2.609035	+/−
15326			♀	50	2.7	53.92	1.58301	WT
15389	B025	ZA94863	♂	82.6	12.3	58.64	7.21149	+/−
15385			♂	84.85	8.85	60.78	5.188755	WT
15391	B025	ZA96588	♂	93.2	12.55	42.29	7.358065	+/−
15390			♂	95.15	10.55	45.98	6.185465	WT

**Table 2 ijms-20-00620-t002:** Hair-related important DEGs between transgenic sheep and wild-type siblings.

Gene Name	Description	Reference
β-catenin	β-Catenin controls hair follicle morphogenesis and stem cell differentiation in the skin	[7]
MMP-7	MMP-7 or matrilysin (β-catenin target gene) is expressed in hair placode keratinocytes	[11]
FN1	β-catenin target gene	[12]
KRT79	Keratin 79 identifies a novel population of migratory epithelial cells that initiate hair canal morphogenesis and regeneration	[13]
KRT2.11	KRT2.11 transcripts are present in wool follicle RNA and are expressed in follicle cortical keratinocytes located above the dermal papilla	[14]
KRT8	KRT8 is an epithelial marker	[15]
KRT71	Mutations in KRT71 are observed in mice, rats, and dogs and are linked to a wavy coat phenotype	[16]
KRT5	KRT5 is a marker of basal and undifferentiated keratinocytes and is increased in the epidermis of alopecic mice	[17]
KRTAP1-1	The ovine KRTAP1-4 gene is clustered with the KRTAP1-1 and KRTAP1-3 genes on chromosome 11 and appears to be associated with wool staple	[18]
KRT25	A homozygous missense variant in type I keratin KRT25 causes autosomal recessive woolly hair	[19]

**Table 3 ijms-20-00620-t003:** Primers for real-time polymerase chain reaction.

Gene		Primer Sequence (5′→3′)	Tm (°C)
β-catenin	F1	AGCGTCGTACATCTATGGG	58
	R1	ATAATCCTGTGGCTTGACC	
KRT25	F7	AACAATATGAGAGCCGAGTA	57
	R7	AACAATATGAGAGCCGAGTA	
KRT71	F8	TCATCGACAAGGTGAGGTTCC	59
	R8	CTGTCCGCCTGTTGATTTCTT	
KRT79	F9	GCAGACATACTCCACCAA	56
	R9	GTTGACCGAGATGCTCTT	
MPC1	F10	GCCATCAATGACATGAAGAA	52
	R10	CACCTTGTAGGCGAATCT	
KRTDAP	F11	GAGGAAGAGACCACCATTG	52
	R11	CGTGCCAGTTCAGGAATT	
ASAP2	F12	CGTTCCTCAAGTTCTCAGT	55
	R12	GCTCCTTCTCCATCTCCT	
ASB7	F13	ACTTAATCGGAGGCTTCAC	55
	R13	GGAGGAACATTCGGCAAT	

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
