# Peer review of "Transcriptome Analysis of Improved Wool Production in Skin-Specific Transgenic Sheep Overexpressing Ovine β-Catenin"

_ijms, 2019, doi:10.3390/ijms20030620_

Round 1

Reviewer 1 Report

The manuscript of Wang et al. is a good study work, in which methods are clear and well explained. The results are consistent with the literature. No additional comments for this work.

Author Response

The comments of the reviewer are appreciated. 

Reviewer 2 Report

In this manuscript entitled " Transcriptome analysis of improved wool production in skin specific Transgenic sheep overexpressing Ovine Beta-catenin", authors have reported the role of increased Beta-catenin on wool production of sheep. Authors, have generated F0 and F1 generation of transgenic sheeps and have attempted to divulge the molecular mechanism of beta catenin role in wool production of sheep. The overall subject of the study is important and provides richness to the sheep wool research. However, authors should address following concerns:

Author's should replace the Fig 1A  with circular full map of vector being used to generate the plasmid of interest, and provide the linear vector map for desired Transgene cassette.

For Fig 1B and 1C, Authors should provide the full image of southern blots. In current form it is hard to interpret the blots especially Fig1c. In Fig 1b plasmid has a very faint band whereas in Fig 1 c plasmid has a bold  band, how do author justify this? Ideally Plasmid band should be stronger and not faint. Have authors also looked for copy number integration from southern blotting? How many integration sites was observed in southern blotting for transgenic sheep in F0 and if thats consistent in F1.

Authors should also provide PCR screening data alongside southern blotting.

Fig 2 Legend starts with " The expression of beta-catenin in the skin of F0 and F1 MICE" it should be sheep not mice. Kindly correct it.

For Figure 2 real time analysis, how did Author's determined the significance? Which statistical tool was used. Same should be described.

Figure 3 and throughout the manuscript change the genotypye symbol (-/-) to WT. It confuses with Knockout of the gene.

Fig 3, use arrow to show inner and outer root sheath. 

Fig 4, Authors describe P<0.01 but fails to describe which statical tool being used to determine significance.

Authors should use term Wild type sibling instead of 'Negative full siblings'. 

Have authors looked for GFP staining in F0 and F1 sheep being generated. If, yes same should be provided in the manuscript.

Authors, should provide the detailed legend for supplementary figure.

Author Response

The comments were highly insightful and enabled us to greatly improve the quality of our manuscript. In the following pages are our point-by-point response to the reviewers’ comments.

Q 1. Arthor’s should replace the Fig1A with circlar full map of vector being used to generate the plasmid of interest, and provide the linear vetor map for desired transgene cassette.

A1: I have replaced Fig. 1A with the complete circular map of the vector in the revised manuscript, and provided the linear vector map in Fig. 1B.

Q2. For Fig1B and 1C, Authors should provide the full image of southern blots, In current form it is hard to interpret the blots especially Fig1c, In Fig1B plasmid has a very faint band whereas in Fig1C plasmid has a bold band, how do author justify this? Ideally plasmid band should be stronger and not faint. Have authors also looked for copy number integration from southern blotting? How many integration sites was observed in southern blotting for transgenic sheep in F0 and if that’s consistent in F1.

A2:

Figure 1 is used to identify positive transgenic sheep. Fig. 1C and 1D were detected by PCR. Fig. 1E and 1F were verification by Southern blot. The aim of these experiments was not to study the number of integration or the number of copies of exogenous fragments.

According to the reviewer comments, we have provided the full images of the Southern blots (Fig. 1 E and 1 F) in the revised manuscript. In this two figures, plasmid template in F0 generation is low than in F1 generation. Further, in order to display the positive signal as far as possible in all the candidate transgenic sheep, the exposure time of Fig. 1F was prolonged, which resulted in the high exposure intensity of the plasmid.

It was unable to calculate the copy number from this Southern blot because the enzyme digestion reaction was purified before hybridization. Sheep skin is thicker and the proportion of keratin and other components is higher than mouse, so Southern blot in sheep genome is relatively not stable. In order to improve the stability of Southern blot in sheep skin genome, we purified the product after the enzyme digestion reaction. The improvements in this technique have significantly improved the efficiency of hybridization. However, it made us unable to quantify the genome copy number in the experiment. We did not design to determine the number of integration sites in this Southern blotting experiment. In order to improve the intensity of hybridization signal, we designed hybridization probe in the middle of the insertion sequence, but not in the flanks of the insertion sequence, which also made it impossible to directly analyze the integration number of the exogenous sequence.

Q3. Authors should also provide PCR screening data alongside southern bloting.

A3: The PCR screening data have been presented in Fig. 1C and 1D in the revised manuscript.

Q4. Fig2 legend starts with”The expression of beta-catenin in the skin of F0 and F1 mice”it should be sheep not mice.

A4: Thank you for pointing this out; I have corrected this accordingly.

Q5. For Figure 2 real time analysis, how did Auther’s determined the significance? Which statistical tool was used, Same should be described.

A5: In this study, we chose full sibs and each sib-pair had the same sex and same month age between case and control group. Therefore, the paired Student’s t-test was used to determine the significance of the difference in data. This has been described in the legend for Fig. 2 in the revised manuscript.

Q6. Figure 3 and throuthout the manuscript change the genotype symbol(-/-) to WT, It confuses with Knockout of the gene.

A6: All such symbols (-/-) have been replaced with “WT” in the revised manuscript.

Q7. Fig3 use arrow to show inner and outer root sheath.

A7: The arrows have been added to Fig. 3 in the revised manuscript.

Q8. Fig4. Authors describe P<0.01, but fails to describe which statical tool being used to determin significance.

A8: I use the paired Student’s t-test to determine the significance; this has been described in the legend for Fig. 4 in the revised manuscript.

Q9. Authors should use term wild type sibling instead of “Negative full siblings”

A9: I have replaced “negative full siblings” with “wild type sibling” throughout the revised manuscript.

Q10. Have authors looked for GFP staining in F0 and F1 sheep being generated. If yes. Same should be provided in the manuscript.

A10: We did not introduce any exogenous protein into the sheep, including GFP, considering the GMO biosafety. Thus, no GFP mRNA was produced in the F0 and F1 transgenic sheep.

Q11. Authors should provide the detailed legend for supplementary figure.

A11: The detailed legend for Fig. S1 has been added below this figure in the revised manuscript.

Sincerely yours,

Xuemei Deng

Round 2

Reviewer 2 Report

Author's have done the commendable job in addressing this reviewer concerns. The manuscript has been improved significantly. No more Comments.